# The Direct-Coupling Method for Analyzing the Performance of Aerostatic Bearings Considering the Fluid–Structure Interaction Effect

**Yangong Wu [1], Wentao Chen [2], Qinghui Zhang [1], Zheng Qiao [1] and Bo Wang [1],***

[1] Center for Precision Engineering, Harbin Institute of Technology, Harbin 150001, China
[2] School of Ocean Engineering, Harbin Institute of Technology (Weihai), Weihai 264200, China
* Correspondence: bradywang@hit.edu.cn

**Abstract:** In the interest of analyzing the effect of the structural deformation root caused by gas pressure on the static features of aerostatic bearings, a fluid–structure interaction (FSI) model based on orifice-type aerostatic bearings is proposed that can predict the characteristics of aerostatic bearings more accurately by using the direct-coupling method (DCM). By using COMSOL Multiphysics, the governing equation matrix of the finite element model of structural deformation and gas film pressure was solved with the integral solution method, and the orifice boundary conditions were calculated with the root iteration method. At the same time, the static performance of I-shaped orifice-type aerostatic bearing with various supply pressures was analyzed theoretically and tested experimentally. The results show that in comparison with the calculation results without taking account of structural deformation, the theoretical values from the model derived in this paper considering the FSI effect are closer to the experimental values. Finally, by using the orthogonal design method, FSI simulation was carried out to analyze how the key dimension factors influence the structural stiffness of the spindle, and it is concluded that the thrust bearing's stiffness is strongly influenced by the thickness of the thrust plate.

**Keywords:** orifice-type aerostatic bearings; fluid–structure interaction model; direct-coupling method (DCM); optimal parameter design

## 1. Introduction

Due to the absence of friction, low heat generation and the good error equalization effect of the gas film, aerostatic bearings provide high rotation precision and low vibration. In ultra-precision machines and precision measuring instruments, aerostatic bearings are commonly used [1–4]. Compared with hydrostatic bearings, the biggest disadvantage of aerostatic bearings is their lower stiffness. As a part of the cutting system, the stiffness of the bearing affects the overall stiffness of the system [5]. Therefore, the most critical technical index for the design of the static gas pressure bearing is its stiffness. However, in the current design of the aerostatic bearing, there is a large difference between the theoretical design values and experimental test results. During the theoretical analysis process, the spindle structure is assumed to be rigid, and the effect of the flexible deformation of the spindle on the bearing performance is not considered. Therefore, when modeling and analyzing aerostatic bearings, the fluid–structure interaction (FSI) effect caused by structural deformation must be fully considered [6].

To date, many scholars have studied the FSI effect of aerostatic bearings. Anton van Beek et al. [7] analyzed how the performance was affected by throttled aerostatic bearings with elastically deformed surfaces by simultaneously solving the 2D Reynolds equation and 3D elastic equation of compressible fluids. Nagendra et al. [8] imported the geometric model into ANSYS Mechanical and conducted Computational Fluid Dynamics (CFD) simulations to analyze the influence of three different materials' bushings on the gas film

pressure distribution. Chen et al. [9] proposed a bearing performance calculation method incorporating the FSI effect and used the gas film thickness after structural deformation to calculate the actual bearing performance. Dhande et al. [10] considered the actual bearing deformation through bidirectional fluid–structure coupling (FSI) and cavitation and concluded that cavitation would reduce the maximum bearing pressure. Based on their proposed comprehensive bidirectional FSI model, Lu et al. [11,12] conducted an investigation into the effects of structural parameters on static bearing performance and the thermal influence of gas film viscosity. Chen et al. [13] established a multiphase flow model for radial bearings that considered both heat and cavitation effects and analyzed the influences of rotational speed, eccentricity and film thickness on the elasto-hydrodynamics of journal bearings with variously shaped grooves. Yan et al. [14] established the FSIW model based on a porous aerostatic bearing and analyzed the FSI deformation law of different porous materials. Wang et al. [15] studied two journal bearings by using the free fluid–structure coupling method (FFSIM); they carried out a bifurcation analysis, studied the influence of rotational speed and rotor mass on the shaft track and generated a phase diagram and frequency response curve. Wei Wang et al. [16] analyzed the effects of gas supply pressure, Young's modulus and porous material thickness on the bearing's static performance. Gao et al. [17,18] proposed a two-wheel optimization design method for the bearing stiffness and key structure size considering FSI. Liu et al. [19] applied the CFD-FSI method to study the dynamic response of the system based on a series of dynamically unbalanced loads and various bearing materials.

The core work of analyzing aerostatic bearings' static performance is mainly to solve the specific Reynolds equation boundary conditions, thus revealing the law of the distribution of the gas film pressure and obtaining the supporting aerostatic bearing performance. However, the Reynolds equation characterized by gas lubrication is a second-order partial differential equation, which makes it difficult to obtain an exact analytical solution. Therefore, the gas lubrication problem is usually solved based on computational fluid dynamics (CFD), while the finite element method is usually used in the analysis of structural deformation. Therefore, when analyzing FSI problems, it is necessary to use different solving modules to calculate the governing equations for the fluid and solid domains and update the boundary conditions in each iteration according to different regional characteristics [20]. However, compared with the monolithic method, due to the non-conservation of energy caused by the calculation lag of the gas domain compared to the solid domain, the above method does not converge at the solid–gas interface [21]. The immersed boundary–lattice Boltzmann method is another Eulerian–Lagrangian approach used for FSI simulations, even in flexible boundary problems [22,23]. In the analysis of FSI problems, the above scholars adopted the sequential solution method to clearly analyze the gas film pressure distribution and structural deformation. However, we find that there are two obvious defects when using the sequential coupling method to solve the FSI problem for aerostatic bearings. Firstly, iteration parameters need to be constantly adjusted in the process of iteration because the calculation lag of the gas domain and solid domain tends to cause the non-convergence of iterations. Secondly, repeated modeling is required when analyzing different structural design parameters. Since the Reynolds equation represents a partial differential equation having a second order, it is also applicable to the finite element method [24]. Therefore, if the elastic deformation equation and Reynolds equation are discretized and formed into a matrix, the accuracy and efficiency of the calculation will be greatly improved.

In this paper, we propose a method for modeling FSI based on the DCM to foresee and understand the static features of aerostatic bearings. In order to verify the accuracy of the theoretical analysis, the static characteristics were tested under different gas supply pressures. In addition, we also investigated how critical structural parameters influence the static performance of thrust bearings, providing the basis for the optimal design.

## 2. DCM-Based FSI Model

### 2.1. Modeling Background

Figure 1a,b show the model of the aerostatic bearing structure. The bearing adopts an I-shaped closed structure, which is composed of upper and lower thrust bearings of different sizes, radial bearings and shaft sleeves with orifice restrictors and air supply pipes. The upper, lower and journal bearings adopt three independent air supply channels. Under working conditions, the high-pressure gas in the gas supply pipeline enters the gas film area through the throttling effect of the restrictor, and after that, it returns to the air through the gas film boundary [25].

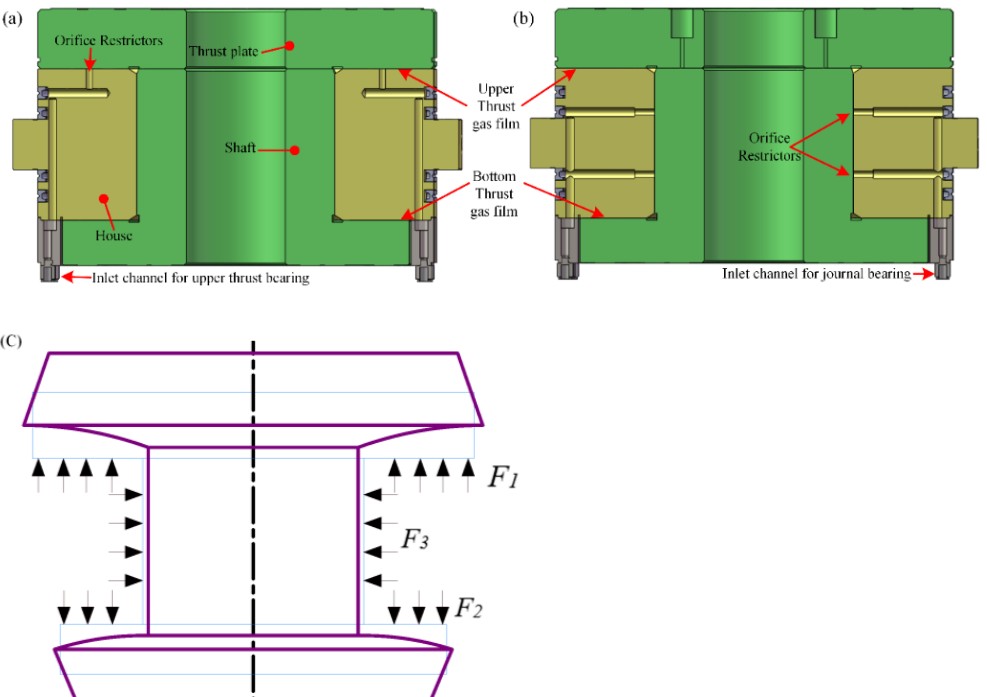

**Figure 1.** FSI model of I-shaped aerostatic bearing: (**a**) the inlet channel for the upper thrust bearing, (**b**) the inlet channel for the journal bearing, (**c**) deformation diagram of the system.

Figure 1c accounts for the interaction between the spindle and gas film under high-pressure external gas. $F_1$ and $F_2$ are the forces exerted by the gas film on two thrust plates, upper and lower, respectively. $F_3$ is the radial gas film force. As a result of the gas film force, the mandrel is mainly stretched by $F_3$. The lower and upper thrust plates are warped by the bending moment under the action of $F_1$ and $F_2$ while being affected by the tension of the mandrel. Finally, the spindle in Figure 1c undergoes structural deformation as a result of the gas film force. At the same time, structural deformation leads to changes in the gap between the spindle and the bushing, which changes the actual thickness and gas film shape, thereby affecting the static performance of the bearing.

### 2.2. The Governing Equations of FSI Systems

As shown in Figure 2, the aerostatic bearing FSI model in the static state is described. $V_f$ and $V_s$ denote the fluid region (gas film) and solid region (spindle), respectively. $S_0$ represents the solid–gas boundary, $S_b$ corresponds to the fixed displacement boundaries, and $S_\sigma$ is the external load application interface. $P$ and $P_0$ are the gas film area pressure and ambient pressure boundary, respectively, and $h$ is the gas film thickness. $n_s$ and $n_f$ are a pair of unit normal vectors with the same size and opposite directions on the solid–gas boundary line, which represent the unit normal vectors outside the solid and fluid boundary elements, respectively.

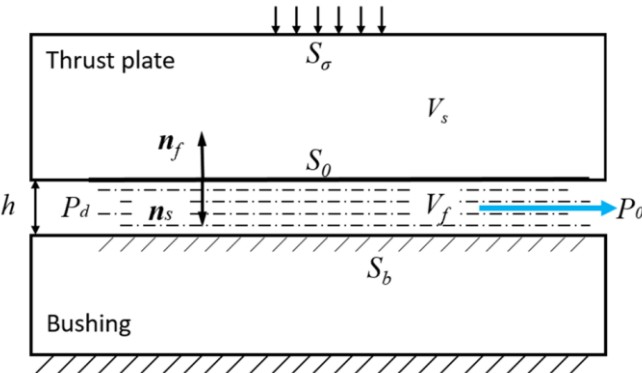

**Figure 2.** The FSI-based model of aerostatic bearing.

To establish a static performance analysis model of the aerostatic bearing considering the influence of structural deformation, it is necessary to conduct an FSI analysis on the two processes of structural deformation and the aerostatic bearing. For the analysis of the gas film region, it is assumed that the gas is an agglutinant and incompressible gas and in a laminar flow state, while the material in the solid region is a linear elastic material. When the main shaft rotates slowly or is stationary, the motion of the gas film region satisfies the static Reynolds equation. For the analysis of solid regions based on elastic mechanics, it is necessary to consider deformation as a variable to establish the static equilibrium differential equation. After introducing the gradient operator into the FSI system, the partial differential control equation is obtained as:

$$\nabla \cdot \boldsymbol{\sigma} + \mathbf{b} = \mathbf{0}$$
$$\nabla \cdot (k \nabla p) = 0 \tag{1}$$

in which $\boldsymbol{\sigma}$ is the solid strain tensor, $\mathbf{b}$ is the body force vector on the surface of the solid region, the pressure of the gas is $p$, and the equation constant is $k = \frac{h^3}{12\eta}$, where $\eta$ and $h$ are the aerodynamic viscosity and the gas film thickness, respectively. Based on the assumption of linear elasticity, the constitutive equation can be further simplified as:

$$\boldsymbol{\sigma} = \mathbf{C} : \boldsymbol{\varepsilon} \tag{2}$$

where $\mathbf{C}$ represents a matrix of linear elastic stiffness, and $\boldsymbol{\varepsilon}$ denotes the strain tensor. In small deformations, the relationship between displacement $\mathbf{u}$ and strain $\boldsymbol{\varepsilon}$ is:

$$\boldsymbol{\varepsilon} = \frac{1}{2}\left[\nabla \mathbf{u} + (\nabla \mathbf{u})^{\mathrm{T}}\right] \tag{3}$$

The boundary conditions at the solid–gas interface should meet the following requirements:

(a) Kinematic boundary conditions: In the gas film region, when the gas enters, the velocity slip phenomenon is ignored; that is, as the gas enters the gas film region, its normal velocity remains continuous. At the same time, it is assumed that the Reynolds equation describes the velocity field in the gas film region. During the static analysis, the velocity at the junction of the solid and gas is set to zero. On the other hand, in the dynamic model, the gas motion must be described by the dynamic Reynolds equation.

(b) Dynamic boundary conditions: The normal force is continuous at the junction of the solid and gas. At the same time, for the solid domain, the following three boundary conditions should also be met:

$$\mathbf{u} = \overline{\mathbf{u}} \text{ on } S_b$$
$$\boldsymbol{\sigma} \cdot \mathbf{n}_s = \overline{\mathbf{T}} \text{ on } S_\sigma \tag{4}$$
$$\boldsymbol{\sigma} \cdot \mathbf{n}_s = -p\mathbf{n}_f \text{ on } S_0$$

where **u** denotes the boundary condition of displacement, and p represents the boundaries of traction. The pressure at the gas boundary should meet the following conditions:

$$p = P_d \text{ in gas chamber}$$
$$p = P_0 \text{ on the edge of atmosphere boundary}$$
(5)

In Equation (5), $P_d$ represents the boundary pressure at the exit of the orifice, and $P_0$ represents the boundary pressure where the gas film meets the atmosphere.

Most of the currently published references on Finite Element Method (FEM) are calculated for either displacement or pressure in a discrete form. In the Appendix A of this paper, using finite element analysis, we establish a discrete linear equation system to directly resolve the FSI issue and provide the finite element solution expression.

### 2.3. The Procedure for Solving FSI Model Based on DCM

From the above analysis, we can see that the structural deformation and gas film pressure are mutually coupled and affect each other. Due to the limited rigidity of the material of the spindle, deformation is caused by the pressure of the gas film. In the case of the spindle structural deformation, the thickness of the gas film will change at the deformed location, thereby affecting the pressure distribution in the gas film. Since structural deformation and film pressure changes occur simultaneously, both processes must be solved simultaneously. At the same time, due to the radial bearing's interaction with its structure and the thrust bearing and gas film, it can be regarded as a nonlinear problem; to solve it, the Newton–Raphson iterative algorithm was used.

Figure 3 shows an overview of the iterative solution process. In this process, COMSOL Multiphysics is used to solve the governing equation of the finite element model by using the global solution method. For each iteration, the initial gas film thickness h is determined after the boundary conditions have been determined, and the influence of the gas film pressure under the gas film thickness on the structural deformation is calculated. If the maximum deformation is greater than 0.1 μm, the deformed gas film thickness is used in iterative calculations until the maximum structural deformation is less than 0.1 μm.

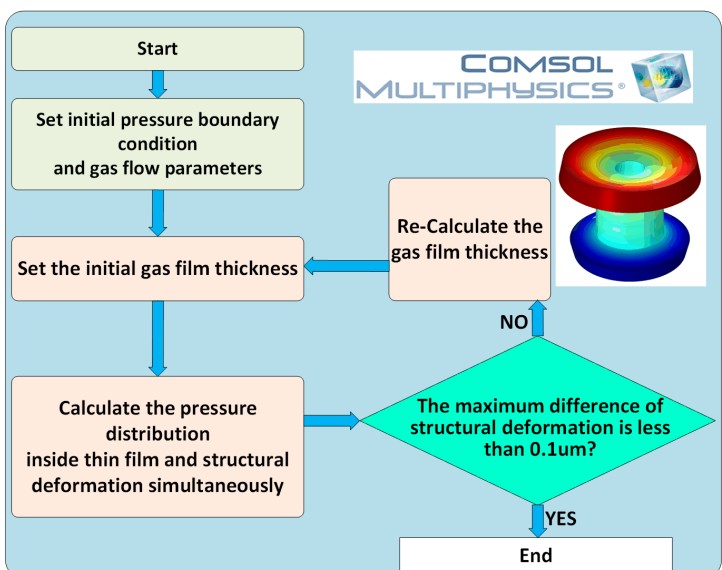

**Figure 3.** The procedure for solving FSI model.

### 2.4. The Calculation Procedure

Figure 4 shows an orifice-type aerostatic thrust bearing, in which an $\varepsilon$-deep pressure-equalizing groove is specially set up to improve rigidity. The high-pressure gas flows into the gas film area from the orifice and finally flows into the surrounding environment from

the thrust plate boundary. Using the finite element method, the Reynolds equation can be solved to calculate the orifice aerostatic thrust bearing's stiffness and the bearing capacity.

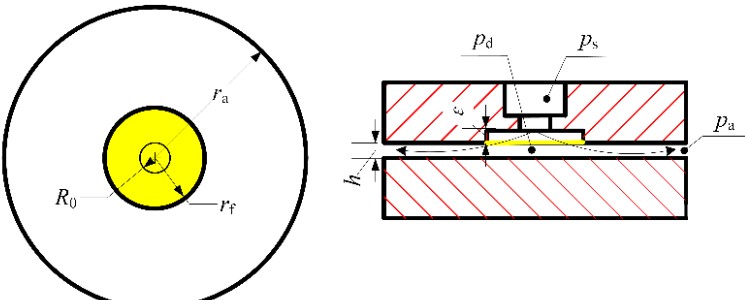

**Figure 4.** Aerostatic bearing with orifice-type diagram.

From Equation (6), it is easy to see that the flow rate is correlated with the pressure drop in the orifice [26]:

$$q = A_0 P_0 \varnothing \sqrt{\frac{2\rho_a}{P_a}} \varphi \qquad (6)$$

where $\varnothing$ represents the restriction coefficient, which is usually 0.8; the area of the orifice restrictor is defined by $A_0$; the supply pressure is defined by $P_0$; the density of the air at standard atmospheric pressure is denoted by $\rho_a$; $P_a$ is standard atmospheric pressure; and the flow function $\varphi$ is [26]:

$$\varphi(\beta) = \begin{cases} [\frac{k}{2}\left(\frac{2}{k+1}\right)^{\frac{k+1}{k-1}}]^{\frac{1}{2}}, \beta \leq \beta_k \\ [\frac{k}{k-1}\left(\beta^{\frac{k}{2}} - \beta^{\frac{k+1}{k}}\right)]^{\frac{1}{2}}, \beta > \beta_k \end{cases} \qquad (7)$$

where k is the specific heat ratio of the gas, $\beta$ is the throttling pressure ratio, and the critical throttle pressure ratio $\beta_k$ is [26]:

$$\beta_k = \left(\frac{2}{k+1}\right)^{\frac{2}{k-1}} \qquad (8)$$

As mentioned above, the FSI model mainly includes the direct-coupling calculation of the aerostatic bearing's boundary conditions and iterative convergence accuracy improvement. Using the direct-coupling calculation method, a FEM-based model was used to simulate the distribution of gas film pressure in the aerostatic bearings and a full-scale spindle structural FEM-based model. The structural deformation of the main shaft and the gas film pressure distribution can be calculated by using COMSOL Multiphysics. However, the establishment of the direct-coupling FSI model mainly describes the characteristics of the gas film region while ignoring the characteristics of the orifice. This is because the orifice characteristics are nonlinear. In this context, the relationship between the flow rate and pressure difference can be examined.

To ensure the convergence of the calculation process, the orifice characteristics of the specific boundary conditions must be correctly characterized. Therefore, this work performed joint calculations in MATLAB and COMSOL and introduced a programmable program to realize nonlinear iteration to characterize the flow balance at the inlet and outlet of the orifice.

As can be seen in Figure 5, the entire calculation process of the proposed FSI model is observed, and the calculation steps are as follows:

1. Firstly, it is important to establish a finite element model for the spindle structure and use the Fluid module of COMSOL to perform preprocessing, such as boundary condition setting and the mesh division of the gas film. As the initial boundary condition for the gas film region, the gas pressure flowing out of the orifice into the pressure-equalizing groove is used, and the contact between the gas film and

the air is used as the outer boundary, where the outer boundary condition is set to 1 bar pressure.

2.  The FEM model adopts the global solution method to solve the pressure distribution, flow and structural deformation of the aerostatic bearing.

3.  In the COMSOL calculation in step (2), the flow rate in the gas film area is obtained, and it is introduced into MATLAB to calculate the post-orifice pressure ($P_{ao}$) of each orifice restrictor; then, the result is compared with the initially given pressure-equalizing groove pressure ($P_{op}$). If the difference between them does not satisfy the convergence condition, $P_{ao}$ is taken as a new $P_{op}$ for a new round of iterations. When $P_{ao}$ and $P_{op}$ meet the convergence conditions, the iterative process ends.

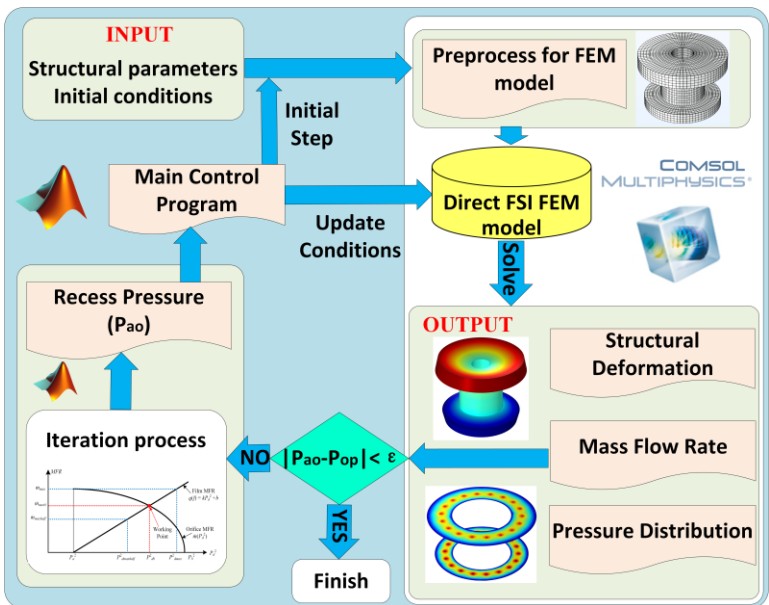

**Figure 5.** The flowchart of the FSI model that was developed in this study.

In order to increase the convergence speed, an effective algorithm must be developed. Currently, the root iteration method (RIM) is often used for aerostatic bearings [20]. In this paper, it is extended to the aerostatic system to improve the iterative calculation speed.

Figure 6 shows the RIM with a single orifice, where the black curve indicates the flow rate through that orifice, calculated by MATLAB according to Equation (6). Equation (1) corresponds to the elastic mechanics expression for the energy expression $1/2\mathbf{f}^T\mathbf{K}\mathbf{f}\text{-}\mathbf{Q}\mathbf{f}^2$, and its corresponding solution is related to the energy extreme. It is known that the film's mass flow rate (MFR) ($q$) is linearly related to the orifice outlet pressure ($P_d{}^2$). Therefore, the pressure $P$ is used as the horizontal coordinate variable, and $q = \mathrm{k}P_d + b$ describes the linear relationship, where k is the slope of the line.

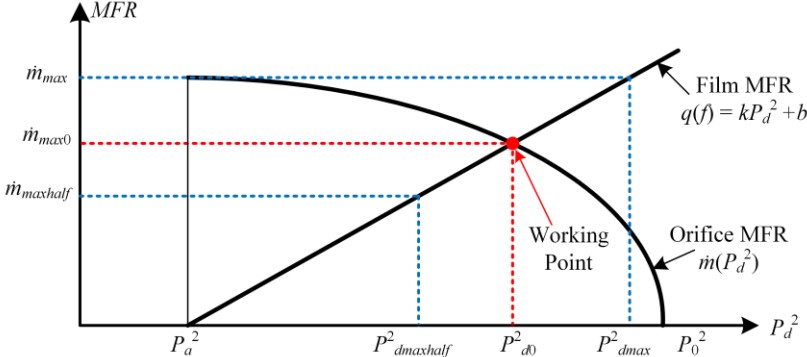

**Figure 6.** RIM schematic of a single orifice.

From the above analysis, it is known that half of the maximum and the maximum MFR ($q_{max}$) that a single orifice can provide are used as boundary conditions. Then, COMSOL can be used to solve the corresponding pressures $P_{dmax}$ and $P_{dmaxhalf}$ after the orifice. Finally, the expression of the gas film pressure distribution can be calculated according to the coordinates of the two points, and by combining the gas film area and the MFR formula at the orifice, the pressure and flow rate of the bearings are calculated as well.

Compared with the solution for gas lubrication with a single orifice, the outlets of multiple orifices are connected to each other through the gas film area, so the effects of deformations of different orifices are all different and affect each other. At the same time, the performance of the bearing is also affected by the change in each restrictor. During one iteration, the MFR for each orifice will be overwritten with new values, which will change the intercept, $b$. Therefore, RIM, as shown in Figure 7, is a more efficient algorithm.

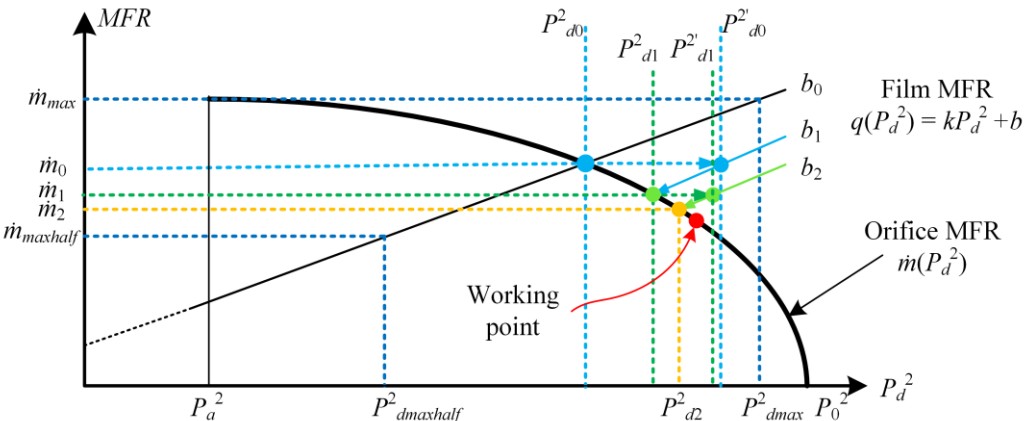

**Figure 7.** RIM schematic of multiple orifices.

Firstly, in the case of a single orifice, the maximum MFR ($q_{max}$) and half of the maximum MFR ($q_{maxhalf}$) that a single orifice can provide are used as boundary conditions. COMSOL can be used to solve the corresponding pressures $P_{dmax}$ and $P_{dmaxhalf}$ after the orifice. The slope k and initial intercept $b_0$ of the Film MFR line can be calculated from the two sets of data. The gas film area and the MFR formula at the orifice are combined to calculate the pressure and flow at the operating point (i.e., $q_0$ and $P_{d0}$).

However, because different orifices interact with each other, with multiple orifices, the $q_0$ and $P_{d0}$ calculated above do not satisfy the Film MFR equation. For this reason, in the initial calculation, we use $q_0$ as the MFR boundary condition and obtain the MFR equation $P_{d0}'$, which does not satisfy the characteristics of an orifice. Therefore, while keeping the slope $k$ constant, we perform iterative calculations based on the new Film MFR equation. Each pair of $q_i$ and $P_{di+1}'$ will get a new intercept $b_{i+1}$ during the iteration. When $b_{i+1}$ is almost unchanged, the iteration ends.

## 3. Simulation Results

According to the spindle structure in Figure 1, its FEM model can be established and meshed. Figure 8a–c show the FEM models with three different mesh densities. By calculating the static performance of the upper thrust bearing with different mesh densities, the calculation accuracy of the algorithm in this paper can be verified.

After running the calculation program, the simulation results considering FSI were obtained, as presented in Figure 9. During the simulation analysis, the gas supply pressure was set to 5.5 Bar, thrust bearings were designed with a gas film thickness of 23 μm on both sides, the designed gas film thickness of the journal bearing was 15 μm, the orifices measured 200 mm in diameter and the rotating shaft weighed 150 kg, and nitride steel was selected as its material.

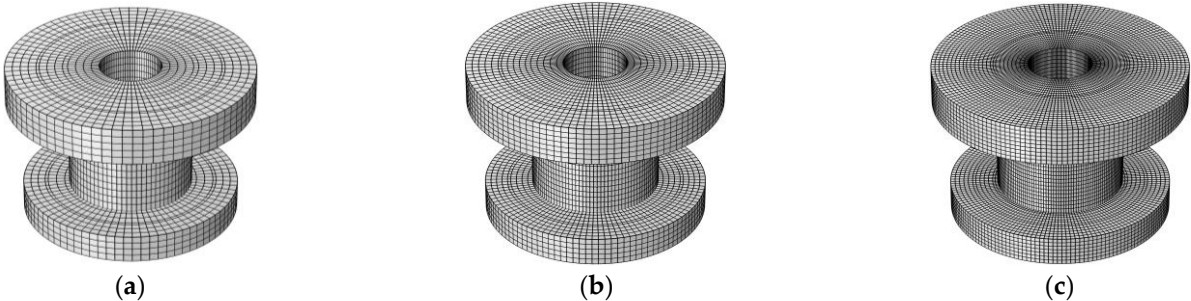

**Figure 8.** FEM meshing with different precisions: (**a**) coarse, (**b**) normal, (**c**) fine.

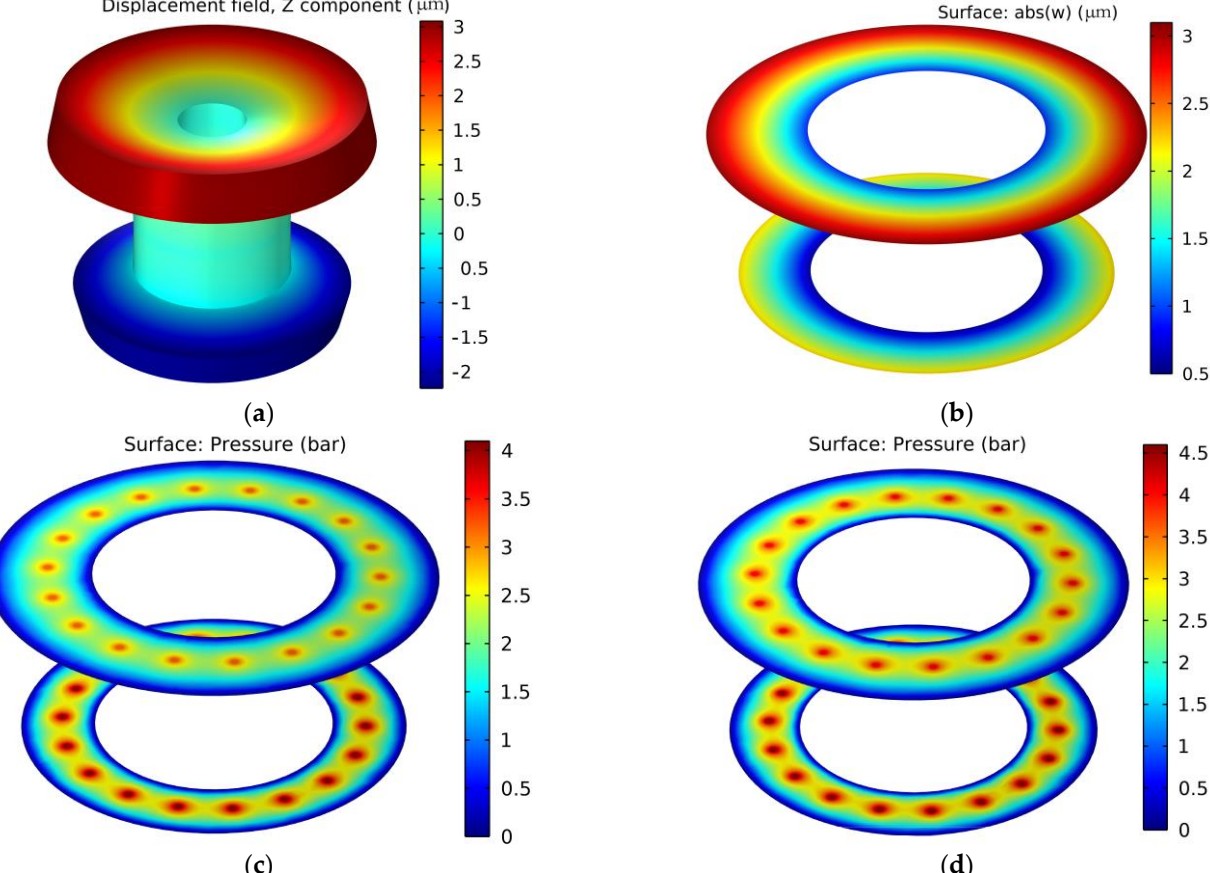

**Figure 9.** Simulation results: (**a**) spindle deformation, (**b**) change in gas film thickness, (**c**) pressure distribution of thrust bearing considering FSI, (**d**) pressure distribution of thrust bearing without considering FSI.

Figure 9a illustrates the thrust plate's deformation after accounting for FSI effects. Under the action of the gas film force, the thrust plate bends and deforms. The deformation range of the upper thrust plate is 0.1–3.3 μm, and the deformation range of the lower thrust plate is 0.1–2.4 μm. It is at the point where the outer diameter of the upper thrust plate meets the air volume that the thrust plate experiences its greatest deformation, and the deformation amount is 3.3 μm. The minimum deformation occurs at the contact of the mandrel of the thrust plate, which is mainly due to the elongation of the shaft under the action of the gas film force, and the deformation is 0.1 μm. As shown in Figure 9b, there is a variation in the thrust bearing's gas film thickness under FSI conditions. Clearly, the change trend of the gas film thickness is the same as that of the thrust plate. At the same time, in a closed thrust bearing, the gas film clearance is the total clearance of the upper and lower thrust bearings; at the contact point between the outer diameter of the lower

thrust plate and the air environment, the gas film thickness will deform the most, and the deformation is 4.8 μm (deformation of the upper thrust plate is 2.7 μm, and bottom thrust deformation is 2.1 μm).

Based on FSI, Figure 9c shows the thrust bearing's pressure distribution. It can be proven that the pressure distribution is the same as that of the traditional orifice-type aerostatic thrust bearing, and the gas film is filled with high-pressure gas from the orifice and finally decays to atmospheric pressure at the gas film boundary. Simultaneously, to analyze the FSI effect on the thrust bearing's performance, we calculated the pressure distribution in the thrust bearing without considering FSI, and Figure 9d illustrates the results. Clearly, the overall pressure of the gas film is greater without considering FSI. This is because after considering the effect of FSI, the gas film gap behind the orifice increases since the thrust plate is bent, and it deforms under the influence of the gas film force, so at the same gas supply pressure, the pressure behind the orifice decreases, leading to a decrease in the overall pressure of the gas film.

The calculation results in Figure 9 demonstrate that the structural deformation of the thrust plate is quite small, so the nonlinear factor can be ignored when modeling the solid phase, and the deformation is assumed to be linear. At the same time, in the case of small deformation, gas wetting still satisfies the assumption of the Reynolds equation.

## 4. Experimental Validation

In order to test the accuracy of the theoretical analysis, we measured the stiffness of the thrust bearing. Under different gas supply pressures, the upper thrust plate deforms, and the thrust bearing flow rate varies. As shown in Figure 10a, it is an orifice-type aerostatic bearing in the experiment. The material of the spindle is nitride steel, the thickness of the upper thrust plate is 60 mm, and the outer diameter is 400 mm. The thickness of the lower thrust plate is 45 mm, and the outer diameter is 350 mm. The outer diameter of the shaft is 200 mm. Gas supply pressure settings are 5, 5.5, 6 and 6.5 bar. In this experiment, two probes were used to measure the stiffness and deformation of the thrust bearing through an inductance micrometer (TESA-TT80 inductance sensor, measuring range ±1 mm, resolution 10 nm). The thrust bearing flow was measured with an SMC-PF2M721-02-D flow meter. Figure 10b is the device used to measure stiffness and deformation. The upper thrust plate's center and edge displacements were measured by using probe A and probe B, respectively. The thrust plate's deflection was determined by comparing the reading values of probe A and probe B.

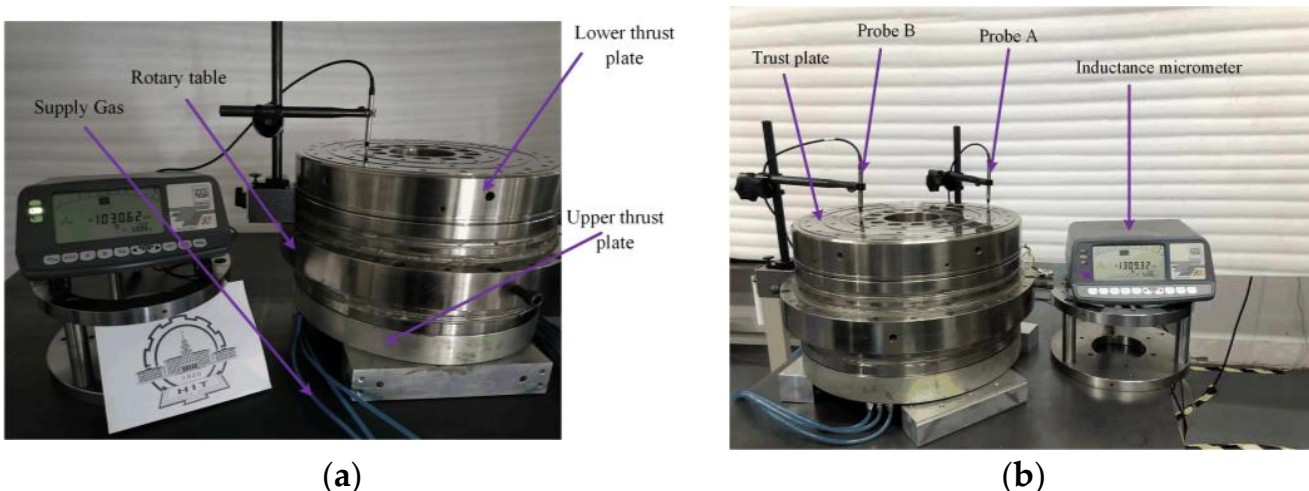

**Figure 10.** Experimental device: (**a**) photo of the aerostatic bearing, (**b**) the measurement device.

Figure 11a shows the static stiffness of the thrust bearing under different gas supply pressures. It can be seen that, without considering the FSI effect, in the case of an increase in gas supply pressure from 5 Bar to 6.5 Bar, the theoretical stiffness of the thrust bearing

improves from 1607 N/μm to 2738 N/μm. According to the results, the thrust bearing's stiffness dropped significantly after considering FSI and only grew from 1293 N/μm to 2073 N/μm as the pressure increased, and the higher the supply pressure, the greater the difference. Clearly, the thickness of the gas film is greatly affected by the structural deformation caused by the gas supply pressure. With an increase in gas supply pressure, the deviation between the actual gas film thickness of the bearing and the ideal value designed by the optimal stiffness criterion increases, and the theoretical stiffness of the thrust bearing decreases correspondingly without considering the FSI effect.

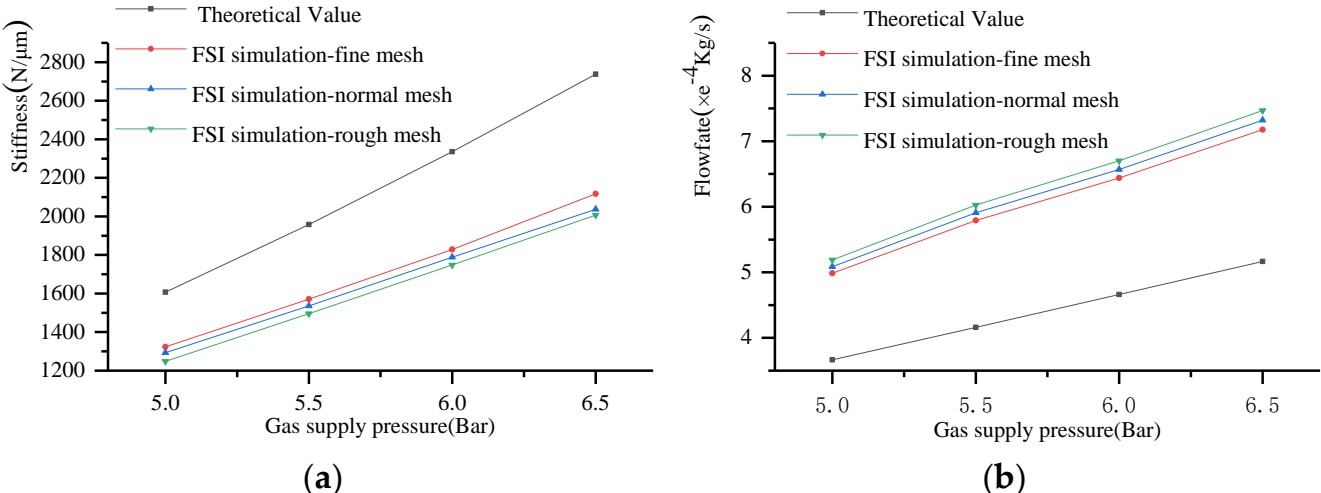

**Figure 11.** Impacts of gas supply pressure: (**a**) stiffness and (**b**) flow rate.

Figure 11b shows the flow of the thrust bearing under different gas supply pressures. The results show that after considering the influence of FSI, the flow rate of the bearing increases significantly, and the deviation increases with the increase in the gas supply pressure. In this case, the thickening of the thrust bearing's gas film is caused by the bending of the thrust plate under the gas film force, thereby increasing the flow rate of the thrust bearing. At the same time, Figure 11 also shows changes in the stiffness and flow of thrust bearings with different grid densities. It can be seen that as the grid density increases, the calculated bearing stiffness and flow will change slightly, but the effect on the overall results is not big. Therefore, the grid density has little influence on the algorithm, which means that a coarser grid can be selected to reduce the calculation time during theoretical analysis.

The comparison of the experimental stiffness of the thrust bearing with the theoretical stiffness considering FSI is given in Figure 12a. It can be seen that the error between the theoretical stiffness and the experimental value is less than 7%, and the calculation accuracy is greatly improved compared with the calculation results without considering FSI. At the same time, with an increase in gas supply pressure, the changing trend of the theoretical stiffness is similar to the experimental value. As shown in Figure 12b, the theoretical analysis and experimental results reveal that the upper thrust plate experienced its maximum deformation. It can be seen that the theoretical calculation error is only 4.3% compared with the experimental value. Figure 12c illustrates the theoretical and experimental flow rates of thrust bearings under varying supply pressures. By comparing the results in Figure 11b with the simulations without the FSI effect, it is evident that the theoretical model derived in this paper can reduce the flow calculation error from 35.3% to 10.7%.

The above results prove that the FSI model established in this paper can more accurately analyze the aerostatic bearing's static performance. On the other hand, it also explains why there is a large error between the calculation results of the traditional theoretical model and the experimental values. However, although the calculation error is greatly

reduced, in actual engineering, it is still affected by non-ideal factors, such as the surface roughness of the gas film and processing and assembly errors.

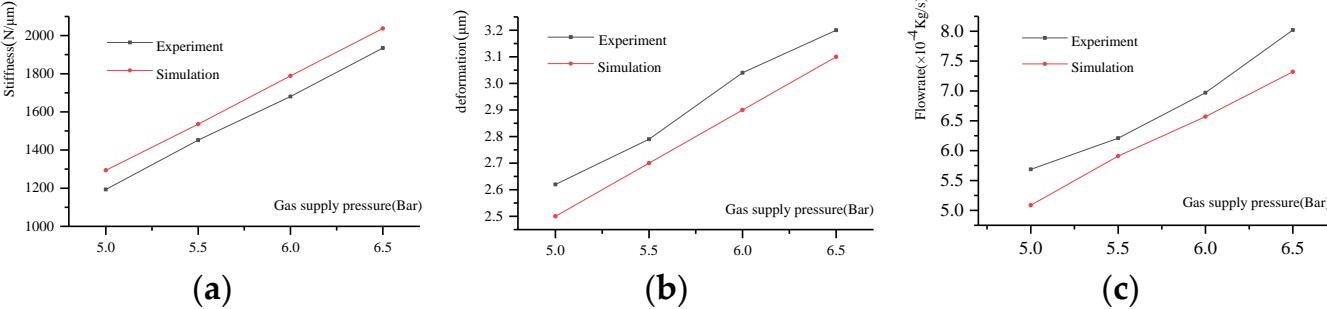

**Figure 12.** Comparison of simulation calculation results with experimental measurements: (**a**) gas pressure–stiffness, (**b**) gas pressure–deformation, (**c**) gas pressure–flow rate.

## 5. Structural Size Optimization Design

Since thrust bearings have a higher load capacity and stiffness index than journal bearings, the non-negligible deformation of the thrust plate is a leading factor in the FSI problem. According to the aforementioned FSI simulation results and experimental verification, it can be seen that the thrust plate is affected by FSI and undergoes bending deformation, which leads to the gas film thickness difference between the design value and the actual value, thereby changing the performance of the thrust bearing. Therefore, structural deformation is coupled with the flow field in the gas film.

The main factors affecting the deformation of the spindle structure are the structural stiffness and gas film forces, which are responsible for determining the thickness of gas films. Among them, the aerostatic bearing is designed based on the gas film force and cannot be changed arbitrarily. Hence, as a means of ensuring the accuracy of the theoretically designed optimal gas film thickness in engineering design, the structural stiffness of the spindle must be increased. Figure 13 shows the critical dimensional factors affecting the spindle stiffness of a typical I-shaped structure. The thrust plate's thickness $H$, the thrust plate's outer diameter $D_1$ and the outer diameter $D_2$ of the mandrel directly determine the overall stiffness of the shaft. FSI simulation was carried out by using the method for orthogonal design, in which three factors correspond to four levels, as shown in Table 1, with the purpose of analyzing the effects of the above parameters on the static performance of the aerostatic bearing. The L16 (43) orthogonal table was chosen to carry out the FSI simulation, and the thrust stiffness K (N/μm) of the aerostatic bearing was used as the target. Orthogonal design parameters and their FSI simulation results are listed in Table 2. The gas supply pressure was set to 5.5 Bar.

The average values of the simulation results in Table 2 were calculated according to the method for same-level range analysis (R), and finally, among the above three structural factors, the performance of the thrust bearing is most affected by the outer diameter of the thrust plate $D_1$, whereas the thrust plate's thickness $H$ has the greatest impact on the performance of the thrust bearing. The thrust bearing's performance is minimally affected. In Figure 14, the relationship between various influencing factors and stiffness is given. Figure 14a shows that the influence of the thrust plate's thickness $H$ on stiffness increases linearly when it is less than 40 mm, and the influence gradually decreases when it is greater than 40 mm. As the thrust plate's thickness increases, its structural rigidity increases, and the deformation of the thrust plate caused by the gas film pressure decreases. Figure 14b shows that the stiffness of the thrust bearing increases linearly as the outer diameter $D_1$ of the thrust plate increases, because the areas with high pressure of the bearing increase as the outer diameter of the thrust bearing increases. At the same time, as shown in Figure 14c, as the outer diameter $D_2$ of the shaft increases, the areas with high pressure on the thrust bearing decrease, resulting in a decrease in the bearing stiffness.

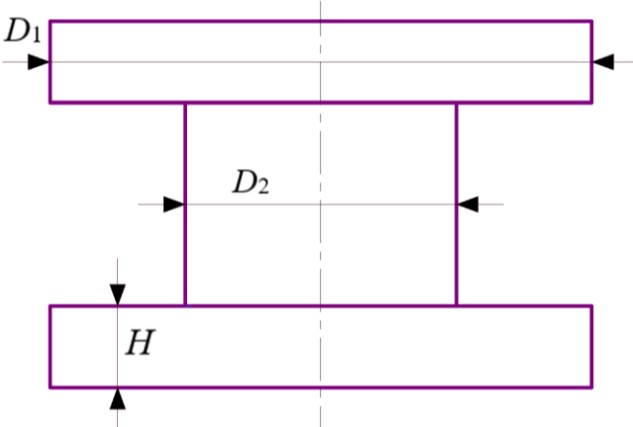

**Figure 13.** The critical dimensional factors affecting the spindle stiffness of a typical I-shaped structure.

**Table 1.** Orthogonal dimension parameters for critical dimensions.

| Levels/Factors | $H$ (mm) | $D_1$ (mm) | $D_2$ (mm) |
|:---:|:---:|:---:|:---:|
| 1 | 30 | 350 | 150 |
| 2 | 45 | 375 | 175 |
| 3 | 60 | 400 | 200 |
| 4 | 75 | 450 | 250 |

**Table 2.** FSI simulation results for orthogonal design parameters.

| Simulation Number | $H$ | $D_1$ | $D_2$ | $K$ |
|:---:|:---:|:---:|:---:|:---:|
| | 1 | 2 | 3 | |
| 1 | 30 | 350 | 150 | 1126 |
| 2 | 30 | 375 | 175 | 1469 |
| 3 | 30 | 400 | 200 | 1713 |
| 4 | 30 | 450 | 250 | 1782 |
| 5 | 45 | 350 | 175 | 1422 |
| 6 | 45 | 375 | 150 | 1600 |
| 7 | 45 | 400 | 250 | 1602 |
| 8 | 45 | 450 | 200 | 2047 |
| 9 | 60 | 350 | 200 | 1319 |
| 10 | 60 | 375 | 250 | 1389 |
| 11 | 60 | 400 | 150 | 1850 |
| 12 | 60 | 450 | 175 | 2143 |
| 13 | 75 | 350 | 250 | 1291 |
| 14 | 75 | 375 | 200 | 1484 |
| 15 | 75 | 400 | 175 | 1717 |
| 16 | 75 | 450 | 150 | 2257 |
| K1 | 1523 | 1290 | 1708 | |
| K2 | 1668 | 1486 | 1688 | |
| K3 | 1675 | 1721 | 1640 | |
| K4 | 1688 | 2057 | 1516 | |
| R | 165 | 767 | 192 | |

Although increasing the thickness of the thrust plate can enhance the stiffness of the thrust bearing, the excessive thickness of the thrust plate will increase the mass and inertia of the spindle, reduce the bearing capacity of the bearing, and cause the deterioration of the dynamic performance of the servo control. In addition, the thrust plate's thickness is also limited by the design space. Therefore, the dimensional parameters of the spindle should

be comprehensively designed according to conditions such as performance and control accuracy requirements.

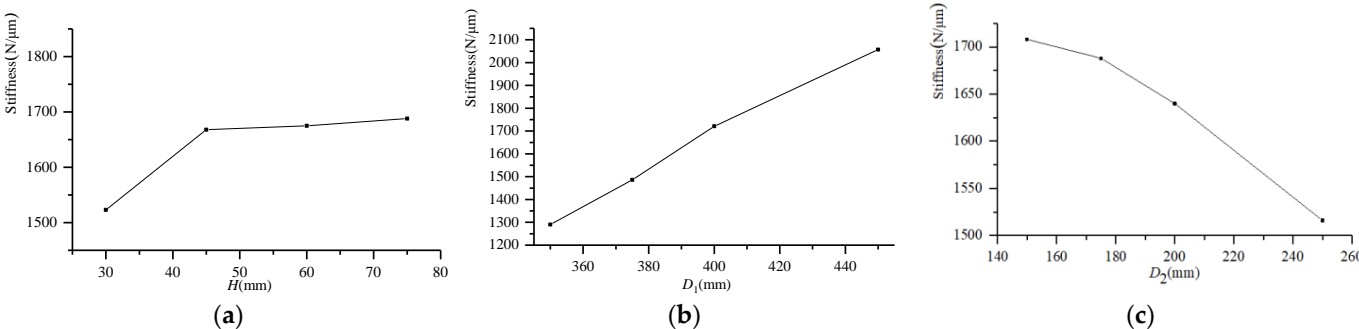

**Figure 14.** Critical dimensional factors affecting stiffness: (**a**) $H$—stiffness, (**b**) $D_1$—deformation, (**c**) $D_2$—flow rate.

## 6. Conclusions

In order to analyze and reveal the influence of fluid–solid coupling on the static gas pressure spindle as a result of the gas film force and its influence on the static performance of the aerostatic bearing, a fluid–solid coupling calculation model was constructed in this study. For the purpose of improving the accuracy of the calculations and the convergence speed, DCM is introduced to solve the gas film flow field and structure field equations, and the numerical solution quickly converges. According to the proposed FSI-based model, the influence of structure size and the effect of supply pressure on the aerostatic bearing's performance were investigated.

The main results achieved are drawn in the following manner:

1. The FSI model established in this work based on the DCM method can greatly improve the calculation accuracy of the aerostatic bearing's performance. The model uses the finite element method to simultaneously calculate the solid field and the fluid field. Compared with the traditional theoretical model, the calculation error is greatly reduced, and at the same time, the method is less affected by the grid division, which greatly reduces the calculation speed. Therefore, compared with the separation approximation method, greater levels of accuracy and efficiency are achieved in the calculation.

2. To establish the FSI model of the I-shaped gas static pressure bearing, the DCM method is proposed in this paper, and how the FSI affects the performance of the thrust bearing under different supply pressures is analyzed based on COMSOL, including deformations in the thrust plate and the stiffness and flow rate of the thrust bearing. Experimental evidence confirms the theoretical analysis, and the results show that the FSI model calculates values that are closer to experimental measurements. Compared with simulation results without taking FSI into account, the variation trend of the stiffness of the thrust bearing in the simulation analysis considering the FSI effect is closer to the experimental results with supply pressure increases.

3. Based on the FSI model, this paper further analyzes the influence of the critical structural dimensions of the I-shaped spindle in terms of the thrust bearing's static performance. According to the results, the thrust plate's thickness has the greatest influence on the thrust stiffness, and the aerostatic bearing design can be optimized on the basis of these results.

In this paper, the FSI model mainly focuses on using the I-shaped orifice-type aerostatic bearing, and in order to optimize the FEM model details, it is necessary to consider the spindle's sensitive positions. At the same time, this method can be extended to other throttling forms of aerostatic bearings, such as porous aerostatic bearings with more obvious effects of fluid–solid coupling problems.

**Author Contributions:** Conceptualization, Y.W., W.C. and B.W.; methodology, W.C. and Z.Q.; software, Y.W. and W.C.; validation, Y.W.; formal analysis, W.C.; investigation, Y.W. and W.C.; resources, Y.W. and Z.Q.; data curation, Y.W. and W.C.; writing—original draft preparation, W.C.; writing—review and editing, Q.Z.; visualization, Q.Z. and W.C.; supervision, Z.Q. and B.W.; project administration, Z.Q. and B.W.; funding acquisition, Z.Q. and B.W. All authors have read and agreed to the published version of the manuscript.

**Funding:** This work was supported by the Open Project Program of the State Key Laboratory of Applied Optics (SKLAO2021001A05); the National Natural Science Foundation of China (No.51905130); and the Heilongjiang Provincial Natural Science Foundation of China (No. LH2020E039).

**Institutional Review Board Statement:** Not applicable.

**Informed Consent Statement:** Not applicable.

**Data Availability Statement:** The data supporting the results reported by the authors can be requested by e-mail.

**Conflicts of Interest:** The authors declare no conflict of interest.

## Appendix A. Discrete Discretization of Partial Differential Governing Equations

Taking the Reynolds equation into account for the gas movement in the gas film field, the gas pressure will not change along the direction of the gas film thickness. We variationally premultiply $\delta\mathbf{u}$ and $\delta p$, respectively, in the governing equations for solid and gas, where $\delta$ represents the variational sign, and after that, integrate the equations governing the system to obtain:

$$\begin{aligned}\int_{\Omega_s} \delta\mathbf{u} \cdot (\nabla \cdot \boldsymbol{\sigma} + \mathbf{b})\mathrm{d}V = 0 \\ \int_{\Omega_f} \delta p[\nabla \cdot (k\nabla p)]\mathrm{d}V = 0\end{aligned} \tag{A1}$$

where $\mathrm{d}V$ I represents volume. $\Omega s$ and $\Omega_f$ represent the solid and gas fields, respectively. Equation (A1) can be simplified to:

$$\begin{aligned}\int_{\Omega_s} \boldsymbol{\sigma} : (\nabla\delta\mathbf{u})\mathrm{d}V - \int_{\Omega_s} \nabla \cdot (\delta\mathbf{u} \cdot \boldsymbol{\sigma})\mathrm{d}V - \int_{\Omega_s} \delta\mathbf{u} \cdot \mathbf{b}\mathrm{d}V = 0 \\ \int_{\Omega_f} (\nabla\delta p) \cdot (k\nabla p)\mathrm{d}V - \int_{\Omega_f} \nabla \cdot [(\delta p)(k\nabla p)]\mathrm{d}V = 0\end{aligned} \tag{A2}$$

Equation (A1) is rewritten according to the Gaussian divergence theorem as:

$$\begin{aligned}\int_{\Omega_s} [\boldsymbol{\sigma} : (\nabla\delta\mathbf{u}) - \delta\mathbf{u} \cdot \mathbf{b}]\mathrm{d}V - \int_{S_\sigma} (\delta\mathbf{u} \cdot \boldsymbol{\sigma}) \cdot \mathbf{n}_s\mathrm{d}S - \int_{S_0} (\delta\mathbf{u} \cdot \boldsymbol{\sigma}) \cdot \mathbf{n}_s\mathrm{d}S - \int_{S_u} (\delta\mathbf{u} \cdot \boldsymbol{\sigma}) \cdot \mathbf{n}_s\mathrm{d}S = 0 \\ \int_{\Omega_f} (\nabla\delta p) \cdot (k\nabla p)\mathrm{d}V - \int_{\partial\Omega_f} [(\delta p)(k\nabla p)] \cdot \mathbf{n}_f\mathrm{d}S = 0\end{aligned} \tag{A3}$$

where $\mathrm{d}S$ represents area.

Depending on the arbitrariness of the variation and the boundary conditions for the pressure, $\delta u = 0$ and $\delta p = 0$ are satisfied at the boundaries $S_u$ and $\partial\Omega_f$. Combined with the elastic mechanics formula, Equation (A3) can be simplified as:

$$\begin{aligned}\int_{\Omega_s} [\boldsymbol{\sigma} : (\delta\boldsymbol{\varepsilon}) - \delta\mathbf{u} \cdot \mathbf{b}]\mathrm{d}V - \int_{S_\sigma} \delta\mathbf{u} \cdot \overline{\mathbf{T}}\mathrm{d}S + \int_{S_0} \delta\mathbf{u} \cdot (p\mathbf{n}_f)\mathrm{d}S = 0 \\ \int_{\Omega_f} (\nabla\delta p) \cdot (k\nabla p)\mathrm{d}V = 0\end{aligned} \tag{A4}$$

Through the above derivation, we obtain equations governing partial differentials in their equivalent integral weak form in the solid and gas fields. Hereafter, the displacement fields of vectors and pressures are discretized using the Voigt symbol, and the discretization of each unit is in the following form:

$$\mathbf{u} = \sum_{i=1}^{m} N_i^{\mathbf{u}}\mathbf{u}_i \quad p = \sum_{i=1}^{n} N_i^{p}p_i \tag{A5}$$

In the solid 0th displacement, $m$ and $n$ are the number of nodes and pressure of the $i$-th node in the solid unit and fluid unit shape functions, respectively.

Similarly, we discretize the stress tensor and pressure gradient:

$$\varepsilon = \sum_{i=1}^{m} \mathbf{B}_i^{\mathbf{u}} \mathbf{u}_i \quad \nabla p = \sum_{i=1}^{n} \mathbf{B}_i^{p} p_i \tag{A6}$$

where $\mathbf{B}_i^{\mathbf{u}}$ and $\mathbf{B}_i^{p}$ are B-matrices at the $i$-th node of the solid and fluid units, respectively. For the 2D case, it follows that

$$\mathbf{B}_i^{\mathbf{u}} = \begin{bmatrix} \partial N_i^{\mathbf{u}}/\partial x & 0 \\ 0 & \partial N_i^{\mathbf{u}}/\partial y \\ \partial N_i^{\mathbf{u}}/\partial y & \partial N_i^{\mathbf{u}}/\partial x \end{bmatrix} \quad \mathbf{B}_i^{p} = \begin{bmatrix} \partial N_i^{p}/\partial x \\ \partial N_i^{p}/\partial y \end{bmatrix}$$

Three-dimensional expressions can also be derived from the above expressions.

At the same time, according to the arbitrariness of the variation, the residual of $\delta\mathbf{u}$ and $\delta p$ can be obtained from the weak form of the equivalent integral:

$$\mathbf{r}_i^{\mathbf{u}} = \int_{\Omega_s} \left(\mathbf{B}_i^{\mathbf{u}}\right)^{\mathrm{T}} \boldsymbol{\sigma} \mathrm{d}V - \int_{\Omega_s} \left(\mathbf{N}_i^{\mathbf{u}}\right)^{\mathrm{T}} \mathbf{b} \mathrm{d}V - \int_{S_\sigma} \left(\mathbf{N}_i^{\mathbf{u}}\right)^{\mathrm{T}} \overline{\mathbf{T}} \mathrm{d}S + \int_{S_0} \left(\mathbf{N}_i^{\mathbf{u}}\right)^{\mathrm{T}} N_j^{p} p_j \mathbf{n}_f \mathrm{d}S$$
$$\mathbf{r}_i^{p} = \int_{\Omega_f} k \left(\mathbf{B}_i^{p}\right)^{\mathrm{T}} \mathbf{B}_j^{p} p_j \mathrm{d}V \tag{A7}$$

$\mathbf{r}^{\mathbf{u}}$ and $\mathbf{r}^{p}$ are nonlinear residual vectors representing elastic solid and fluid medium deformations and pressure fields. For the purpose of solving the nonlinear problem in the iterative process, the algorithm utilizes Newton–Raphson iteration, and the tangent stiffness matrix is calculated as:

$$\mathbf{K}_{ij}^{\mathbf{uu}} = \frac{\partial \mathbf{r}_i^{\mathbf{u}}}{\partial \mathbf{u}_j} = \int_{\Omega_s} \left(\mathbf{B}_i^{\mathbf{u}}\right)^{\mathrm{T}} \mathbf{C} \, \mathbf{B}_j^{\mathbf{u}} \mathrm{d}V$$
$$\mathbf{K}_{ij}^{\mathbf{u}p} = \frac{\partial \mathbf{r}_i^{\mathbf{u}}}{\partial p_j} = \frac{1}{\rho_f} \int_{S_0} \rho_f \left(\mathbf{N}_i^{\mathbf{u}}\right)^{\mathrm{T}} \mathbf{n}_f N_j^{p} \mathrm{d}S \tag{A8a}$$

$$\mathbf{K}_{ij}^{p\mathbf{u}} = \frac{\partial \mathbf{r}_i^{p}}{\partial \mathbf{u}_j} = \mathbf{0}$$
$$\mathbf{K}_{ij}^{pp} = \frac{\partial \mathbf{r}_i^{p}}{\partial p_j} = \int_{\Omega_f} k \left(\mathbf{B}_i^{p}\right)^{\mathrm{T}} \mathbf{B}_j^{p} \mathrm{d}V \tag{A8b}$$

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
