# Peer review of "The Direct-Coupling Method for Analyzing the Performance of Aerostatic Bearings Considering the Fluid–Structure Interaction Effect"

_lubricants, doi:10.3390/lubricants11030148_

Round 1

Reviewer 1 Report

Manuscript ID: lubricants-2246783

Title: The direct-coupling method analyzing the performance of aerostatic bearings considering fluid-structure interaction effect

The authores present an an Fluid-solid interaction modeling method based on the irect coupling method to predict and analyze the static performance of aerostatic bearings. The subject addressed in this article is worthy of investigation. Also, the conclusions are supported by the data and organization of the manuscript is appropriate. The authors should address the following comments before the final decision:

  ·         Enrich abstract and conclusion sections of the paper with more quantitative data from the research model

·         In line 76, the DCM abbreviation is not appropriate.

·         Line 111-119 the format of the text is not correct.

·         The governing equation should be referred to.

·         Section 3: A mesh grid study should be presented for this numerical simulation.

·         Section 4: Be more careful about using the word "validation" and “verification”.  Please see the following reference for more information:

ASME, Standard for Verification and Validation in Computational Fluid Dynamics and Heat Transfer, ASME     V & V, 20-2009, 2009, New York: American Society of Mechanical Engineers.

·         Immersed boundary-lattice Boltzmann method is another Eulerian-Lagrangian approach used for the FSI  simulation even in flexible boundary problems. It is suggested to introduce this method in literature review for more information of the readers. The following papers will be useful: 10.1016/j.molliq.2020.114941; 10.1016/j.oceaneng.2022.111025

Reviewer 2 Report

The paper is written well and brings good results. I recommend it for publishing after minor editing. The following formal errors should be corrected:

Line 11: Missing space before parenthesis and after period. 

Line 76: Letter "d" in word "due" is in a different font.

Line 79: Missing space after period.

Line 161: It is mentioned that b represents the traction boundary condition. But there is no letter "b" in the respective equations.

Equation 5: Different font from other equations.

Figures: Some labels begin with "Figure", while others with "Fig." This should be unified.

Lines 238239: Should be "When p_a0 and p_op meet the convergence conditions..." instead of "Until..."

Line 251: It should be Pd^2 instead of Pd.

Line 278: The end of sentence "that does not satisfy the orifice" makes no sense for me.

Line 280: "Until" should be replaced e.g. by "When".

Line 291: "kg" is in a different font.

Lines 315317: Different font used.

Line 336: Should probably be singular "bearing" instead of plural "bearings".

Line 398: The end of the sentence should be reformulated.

Table 1: Labels should be formatted as in Table 2 (using italics and subscripts where appropriate).

Lines 465466: The same is repeated in the following lines; therefore, this should be removed.

Line 476: "performance of the thrust bearing" instead of "on"

Lines 480484: The sentence should be reformulated. It can hardly be understood.
